# Semi-Targeted Profiling of the Lipidome Changes Induced by *Erysiphe Necator* in Disease-Resistant and *Vitis vinifera* L. Varieties

**DOI:** 10.3390/ijms24044072

**Published:** 2023-02-17

**Authors:** Ramona Mihaela Ciubotaru, Mar Garcia-Aloy, Domenico Masuero, Pietro Franceschi, Luca Zulini, Marco Stefanini, Michael Oberhuber, Peter Robatscher, Giulia Chitarrini, Urska Vrhovsek

**Affiliations:** 1Department of Agri-Food, Environmental and Animal Sciences, University of Udine, Via delle Scienze 206, 33100 Udine, Italy; 2Food Quality and Nutrition Department, Research and Innovation Centre, Fondazione Edmund Mach, Via Mach 1, 38089 San Michele all’Adige, Italy; 3Unit of Computational Biology, Research and Innovation Centre, Fondazione Edmund Mach, Via Mach 1, 38089 San Michele all’Adige, Italy; 4Genomics and Biology of Fruit Crops Department, Research and Innovation Centre, Fondazione Edmund Mach, Via Mach 1, 38089 San Michele all’Adige, Italy; 5Laboratory for Flavours and Metabolites, Laimburg Research Centre, Laimburg 6, Pfatten (Vadena), 39040 Auer, Italy

**Keywords:** *Vitis vinifera*, resistant varieties, plant lipid metabolism, powdery mildew, biomarkers

## Abstract

The ascomycete *Erysiphe necator* is a serious pathogen in viticulture. Despite the fact that some grapevine genotypes exhibit mono-locus or pyramided resistance to this fungus, the lipidomics basis of these genotypes’ defense mechanisms remains unknown. Lipid molecules have critical functions in plant defenses, acting as structural barriers in the cell wall that limit pathogen access or as signaling molecules after stress responses that may regulate innate plant immunity. To unravel and better understand their involvement in plant defense, we used a novel approach of ultra-high performance liquid chromatography (UHPLC)-MS/MS to study how *E. necator* infection changes the lipid profile of genotypes with different sources of resistance, including BC4 (*Run1*), “Kishmish vatkhana” (*Ren1*), F26P92 (*Ren3*; *Ren9*), and “Teroldego” (a susceptible genotype), at 0, 24, and 48 hpi. The lipidome alterations were most visible at 24 hpi for BC4 and F26P92, and at 48 hpi for “Kishmish vatkhana”. Among the most abundant lipids in grapevine leaves were the extra-plastidial lipids: glycerophosphocholine (PCs), glycerophosphoethanolamine (PEs) and the signaling lipids: glycerophosphates (Pas) and glycerophosphoinositols (PIs), followed by the plastid lipids: glycerophosphoglycerols (PGs), monogalactosyldiacylglycerols (MGDGs), and digalactosyldiacylglycerols (DGDGs) and, in lower amounts lyso-glycerophosphocholines (LPCs), lyso-glycerophosphoglycerols (LPGs), lyso-glycerophosphoinositols (LPIs), and lyso-glycerophosphoethanolamine (LPEs). Furthermore, the three resistant genotypes had the most prevalent down-accumulated lipid classes, while the susceptible genotype had the most prevalent up-accumulated lipid classes.

## 1. Introduction

Lipids are essential plant components. The lipidome is the whole lipid profile of an organism, tissue, or cell [1], and lipidomics is the detailed study of lipid molecules, including identification, quantification, and understanding of their significance in biological systems [1,2]. LIPID MAPS (https://www.lipidmaps.org (accessed on 10 November 2022)) classifies lipids into separate categories based on the distinct hydrophilic and hydrophobic constituents that form the lipid. Fatty acyls (FAs), glycerolipids (GLs), glycerophospholipids (GPs), sphingolipids (SPs), saccharolipids (SLs), polyketides (PKs), sterol lipids (STs), and prenol lipids (PRs) are the eight major categories and can be identified by their chemically functional backbone structures [3]. In plants, they perform a variety of roles, including those related to cell architecture [4], energy storage [5], cell signaling [6], reducing stress tolerance [7], and symbiotic and pathogenic relationships [8].

In the interaction between pathogens and plants, lipids are crucial, particularly in the following three key areas: pathogen development and life cycle completion, pathogen recognition and host-initiated defense response, and impeding host defense mechanisms to overcome resistance [9]. As has been proven several times, lipids play an important role in both types of plant immunity, pathogen-associated molecular pattern (PAMP)-triggered immunity (PTI) [10] and effector-triggered immunity (ETI) [11]. When pathogens enter the host, the cuticle is the first barrier they meet. Pathogens penetrate plant tissue and encounter the apoplast, one of the most important cellular compartments in the defense response. Here, pathogens secrete molecular effectors during plant–microbe interactions, generating a wide range of changes in this compartment [12], with still-unknown effects on the modulation of lipids [13]. Nonetheless, there is little evidence of the relevance of extracellular lipids in plant–pathogen interactions in the creation of systemic acquired resistance (SAR) [14].

It is known that upon pathogen interaction, a plant’s lipidic profile may experience changes frequently linked to the modulation of membrane fluidity and enzymatic and non-enzymatic creation of bioactive lipid mediators such as oxylipins, FA oxidation products, and lipids [15]. This modulation has been identified as a critical element in triggering plant immunity [16,17,18]. Although structural lipids derived from primary metabolism function in order to restrict pathogen penetration, infections caused by pathogens such as *Erysiphe necator* can overcome the basal defensive systems in many economically important grapevine cultivars. The disease can be difficult to detect, especially in the early stages, as signs and symptoms are often subtle. Failure to prevent and/or control powdery mildew often results in insufficient fungicide spray coverage, and because the majority of these fungicides are site-specific, recurrent application results in fungicide-resistant isolates [19]. Thus, valorizing resistant cultivars with resistance quantitative trait loci (QTLs) named *Ren* and *Run* (conferring resistance to *Erysiphe necator* and *Uncinula necator*, respectively) is the most promising technique for reducing chemical use in viticulture and avoiding the establishment of *E. necator* resistance isolates [19,20]. However, it must be highlighted that using varieties with only one gene or locus can encourage the selection of fungal isolates capable of overcoming these key resistance loci [21]. To avoid such resistance breakdowns, a different approach is to employ pyramided cultivars, which store many resistant genes/loci against the same pathogen/disease [22].

We previously provided metabolomics evidence on the early interaction between grapevine varieties with one locus and grapevine varieties with several loci and *E. necator* [23]. We discovered that the class of molecules most affected by the pathogen was lipids, highlighting the importance of lipids in grapevine defense against the powdery mildew causative agent. The increased accumulation in the plant metabolome of four fatty acids (behenic acid, palmitoleic acid, arachidic acid, and oleic acid+*cis* vaccenic) and one prenol (oleanolic acid) showed their involvement in plant defense mechanisms. Despite this evidence and a growing interest in the involvement of lipids and lipid-related compounds in plant–pathogen interactions, few studies have focused on the interaction of lipids with grapevine diseases. The grapevine leaf–*Plasmopara viticola* pathosystem has received the most attention [16,24,25,26,27], whereas the interaction between *E. necator* and grapevine leaf lipids has only been reported in one untargeted metabolomics study [28]. In general, lipidomics research is needed to better understand plant defense mechanisms against *E. necator*, particularly the role of lipids in regulating plant defense responses in *E. necator*-affected mono-locus and pyramided grapevine genotypes.

Thus, we decided to extend our previous investigation on *E. necator* and focus solely on the changes brought about by the pathogen in the plant lipidome. We did so by using a newly developed sensitive and accurate semi-targeted ultra-high performance liquid chromatography (UHPLC)-MS/MS method [3]. This allowed us to acquire a more holistic picture due to its power in analyzing and quantifying a vast number of chemical compounds from multiple classes of lipids in a single analytical run, as opposed to the earlier employed targeted method of [26], which considered only 32 lipid compounds. For this purpose, we studied three of the previously investigated resistant grapevine varieties with a different percentage of lipids modulated as a reaction to the infection with the pathogen *E. necator*, and screened them for two years to detect changes in the lipid profile during plant–pathogen interactions. In this work, the lack of knowledge on the impact of *E. necator* on the lipidome of grapevine leaves was addressed for the first time. This brought us closer to understanding grapevine lipid-mediated defense mechanisms and highlighted potential compounds for future disease tolerance/resistance breeding initiatives.

## 2. Results

We investigated 8098 lipids of possible interest for grapevine defense using the semi-targeted ultra-high performance liquid chromatography (UHPLC)-MS/MS approach. Among the investigated lipids, 271 were detected within the inoculated and non-inoculated leaves (control) belonging to the four chemical categories studied (glycerophospholipids, glycerolipids, sphingolipids, and fatty acids). Appendix A (sheet 3) shows the semi-quantification of all detected lipids expressed as µg/g of fresh leaf powder for each genotype in both years.

### 2.1. Phenotypic Resistance

The four genotypes studied scored differently on the scale of the Organisation Internationale de la Vigne et du Vin (OIV-455 descriptors). At 7 dpi (days post-inoculation), we attributed an OIV-455 score of 9 to the genotype with total resistance (BC4), an OIV-455 score of 7 to the two genotypes with partial resistance (“Kishmish vatkana”, and F26P92), and an OIV-455 score of 1 to the susceptible genotype “Teroldego”. Appendix A contains the OIV-455 scores assigned for grapevine leaf resistance to powdery mildew.

### 2.2. Lipid Modulation of the Grapevine–E. necator Interaction during the First Hours of Infection

We focused on the lipidome modifications of the grapevine leaves in response to the artificial infection at the time points of 24 and 48 hpi (hours post-inoculation), taking into account that at 0 hpi, the plant lipidome should not suffer any change after the effect of the year was removed.

Out of 271 lipids identified and semi-quantified, the percentage of lipids within their corresponding class that showed a significant modulation is shown in Figure 1. The dots contained within the vertical green line represent the percentages of lipid modulation at 24 hpi, whereas the ones within the red line represent the percentages of lipid modulation at 48 hpi. The most modulated lipid classes were identified at 24 hpi in the resistant genotypes BC4 (13 classes with 55 modulated lipids) and F26P92 (13 classes with 69 modulated lipids). By 48 hpi, however, both BC4 and F26P92 showed a decreased response (10 classes with 33 modulated lipids and 8 classes with 11 modulated lipids, respectively). Interestingly, “Kishmish vatkhana” displayed a different behavior than the other resistant varieties. It showed a low level of lipid modulation with only 3 modulated lipids belonging to 3 different classes at 24 hpi, which increased to 15 modulated lipids of 8 classes at 48 hpi. The susceptible genotype “Teroldego” modulated 13 lipids from 7 classes at 24 hpi, which then increased to 100 modulated lipids from 11 classes at 48 hpi (Figure 1).

To go deeper into the molecular aspects of the modulation, the previous results were further explored in a series of volcano plots, as presented in Figure 2 and Figure 3. The figures emphasize all the classes of lipids (in gray) and highlight each class of modulated lipids with a different color (independently of their statistical significance) for each genotype. The discontinued horizontal red line represented in the graph indicates the threshold for statistical significance (uncorrected *p* < 0.05), whereas the discontinued vertical green lines were used to select strongly reacting lipids (absolute d > 1). The lipids situated on the right of the discontinued vertical green line indicate that infected plants produced more lipids (up-accumulation). Consequently, a high tail on the right arm of the volcano denotes a positive metabolic response to infection. On the other hand, the lipids above the threshold situated on the left of the discontinued vertical green line indicate that infected plants produced fewer lipids (down-accumulation). The reduced level of lipids in response to infection appears as the high tail of the volcano’s left arm. The modulated lipids, both up-accumulated and down-accumulated, with their calculated effect size and *p*-values, are listed in Appendix A (24 hpi in sheet 1 and 48 hpi in sheet 2).

The genotype BC4 displayed 32 lipid compounds that were up-accumulated and 23 lipid compounds that were down-accumulated at 24 hpi. The most prevalent up-accumulated compounds were glycerophospholipids in the PE, PG, and PI classes, whereas the most prevalent down-accumulated compounds were glycerolipids in the DGDG class (Figure 2). At 48 hpi, only 4 lipids showed up-accumulation, whereas 29 lipids were down-accumulated with the most prevalent modulation being the down-accumulation of the PE and PC classes (Figure 3).

At 24 hpi, F26P92 had up-accumulated 1 lipid compound from the glycerophospholipids in the LPC class and 1 sphingolipid from the dhCER class, while down-accumulating 67 lipid compounds. Among these, the most prevalent compounds were the glycerophospholipids (19 lipid compounds in PA and 6 lipid compounds in the PE class) and glycerolipids (20 lipid compounds in the DGDG class and 11 lipid compounds in the MGDG class) (Figure 2). It is interesting to note that at 48 hpi, there was a decrease in the number of lipid compounds that were down-accumulated (seven), which included glycerophospholipids and glycerolipids, and a slight increase in the number of lipid compounds that were up-accumulated (four), which included glycerophospholipids, glycerolipids, and fatty acids (Figure 3).

At 24 hpi, the genotype “Kishmish vatkhana” had one down-accumulated compound belonging to glycerolipids (MGDGs) and one component up-accumulated belonging to glycerophospholipids (PEs) (Figure 2). At 48 hpi, a different pattern of behavior was discerned, with 10 lipid compounds up-accumulated—the most prevalent being in the glycerolipid (MGDG and DGDG) and glycerophospholipid (PC) classes—and 5 lipid compounds down-accumulated, each one belonging to a different class of glycerophospholipids and fatty acids (Figure 3).

“Teroldego” displayed at 24 hpi 5 down-accumulated lipid compounds and 8 up-accumulated lipid compounds in the glycerophospholipid and glycerolipid groups (Figure 2), whereas, at 48 hpi, there were 10 down-accumulated lipid compounds in the glycerophospholipid groups and a significant increase in the up-accumulated lipid compounds (90). The most prevalent up-accumulated compounds were the glycerophospholipids (30 lipid compounds in PE group and 26 lipid compounds in PC group) and the glycerolipids (13 lipid compounds in the DGDG class and 9 lipid compounds in MGDG) (Figure 3).

## 3. Discussion

In this study, we investigated how the lipidome of grapevine leaf tissue can be impacted by *E. necator*. To our knowledge, this work is the first to describe how lipid metabolism is modulated in the leaves of two mono-locus resistant and one pyramided resistant *V. vinifera* varieties compared to a susceptible variety upon *E. necator* infection.

The results of our study show that modulated lipids can be detected in *E. necator*-infected tissues at very early stages (24 and 48 hpi) of the infection process. Furthermore, the findings of our investigation reveal a distinct percentage of modulation of lipids in the first hours following *E. necator* artificial infection between the susceptible and the resistant genotypes. According to a study by [29], the development of the pathogen’s infectious structure, which takes around 24 h [30], is the only time when defense metabolites are induced and accumulated more. This was observed in the resistant genotypes BC4 and F26P92, which had the strongest modulation of several lipid classes at 24 hpi, followed by a lower modulation of some classes at 48 hpi. In contrast, the resistant genotype “Kishmish vatkhana” seemed to have a more limited modulation at 24 hpi and an increase in the lipid class modulation at 48 hpi, whereas “Teroldego” showed a high modulation of lipids, particularly at 48 hpi. These results are in accordance with the previous studies [27,31], which were carried out on the pathosystem grapevine—*P. viticola*—and showed that the plant defense mechanism was fully engaged in the first 48 h after infection.

In this work, the differing lipid modulation levels observed between genotypes as a result of the *E. necator* infection could be attributed in part to the genotype and phenotype, which have a role in influencing plant lipid abundance [26]. In fact, at a genetic level, the presence of multiple resistance loci does not necessarily result in a higher resistance response for all genotypes [32,33], indicating that combinations of loci such as *Ren3Ren9* do not always have additive effects [20,34]. This result was observed with the genotypes F26P92 and BC4. In this case, the two genotypes showed similar levels of lipid modulation despite the fact that F26P92 has two resistant loci (*Ren3* and *Ren9*) and BC4 is a mono-locus genotype resistant only through *Run1*. “Kishmish vatkhana” is likewise a mono-locus genotype resistant through *Ren1*; however, it showed a more limited lipid modulation than BC4 and F26P92, which confirms the role of the genetic influence in plant lipid modulation. Moreover, the different genotypes had different phenotypic responses to the pathogen. When there are considerably suppressed symptoms or no detectable symptoms of infection at all, the level of resistance is referred to as “total”, and when there is a decrease in symptoms but no complete disappearance, the level of resistance is referred to as “partial” [35,36]. The OIV-455 descriptors indicated BC4 as a genotype with very high resistance, which is in accordance with the studies of [20,37], which classified BC4 as a genotype with total resistance. Ref. [20] found that varieties carrying *Run1* the locus, such as BC4, have a quick HR that could be observed at 48 hpi in cells where the fungus developed secondary hyphae, as evidenced by the rise in ROSs (reactive oxygen species) and the appearance of PCD (programmed cell death). The buildup of callose deposits at the *E. necator* infection site is another reaction caused by *Run1*. The genotypes “Kishmish vatkhana” and F26P92 were characterized through the OIV-455 descriptors as having a high resistance, which corroborates the partial resistance found in the literature for these two genotypes [34,38,39,40]. Ref. [20] found that the fungus attacked 84% fewer cells in varieties that carry the *Ren1* locus, such as “Kishmish vatkhana”. Other reactions include the stimulation of ROSs at 96 hpi, the induction of PCD at 48 hpi, and the growth of callose deposits. Ref. [34] found similar strong resistance responses for varieties that carry the two *Ren3Ren9* loci, such as F26P92. Therefore, the loci’s level of resistance (whether total or partial) seems to be more significant than the overall number of loci present in the genotypes [35].

The modulation observed in the susceptible genotype may be due to a late response of the plants to the infection that could have become stronger at 48 hpi. This modulation could indicate the start of a basal defense similar to the response in resistant plants but insufficient in timing and/or intensity to stop the spread of the disease [41]. Moreover, the OIV-455 descriptors classified “Teroldego” as a genotype with very low resistance, as seen in our phenotypic evaluation at 7 dpi, which could be predicted given its susceptibility to the pathogen.

The most important changes seen in the lipidome of the investigated genotypes are the up-accumulation and down-accumulation of lipids as a response to *E. necator* infection. The most prevalent classes of lipids in the resistant genotypes were primarily down-accumulated, whereas the most prevalent classes of lipids in the susceptible “Teroldego” were primarily up-accumulated.

An exception to this observation for resistant genotypes is BC4 at 24 hpi, which had up-accumulated lipids mainly from glycerophospholipids in the PE (glycerophosphoethanolamine), PI (glycerophosphoinositol), and PG (glycerophosphoglycerol) classes. Understanding lipid alterations helps us understand how cells operate, because glycerophospholipids make up the majority of the cellular membrane [26]. PG is a thylakoid lipid with an important role in photosynthesis [42]. PI is produced by phosphatases and lipid kinases, and as a signaling lipid, it serves as a precursor for stress-signaling lipids such as DAG (diacylglycerol) and inositol phosphatases [43]. Together with PE, an extra-plastidial lipid, they are major membrane lipids that play a crucial role in transporting materials and maintaining the structure of cell plants. As a consequence, the up-accumulation of the PG, PE, and PI lipid classes in this genotype during the first 24 h of infection may indicate the plant’s struggle to overcome stress brought on by the infection. Thus, it may produce more lipids that could regulate cell photosynthesis as in normal circumstances and activate phospholipids as a barrier to protect the cell walls at the extracellular signal perception of the pathogen. Interestingly, the extra-plastidial lipid PE is down-accumulated in BC4 at 48 hpi together with PCs; a down-accumulation of the PE is seen also in F26P92 at 24 hpi, whereas for “Teroldego”, both PEs and PCs are up-accumulated. Similar results were found in the study of [16]. After *P. viticola* inoculation, the resistant grapevine genotype “Regent” showed a tendency to have a decrease in PE and PC content, while the susceptible grapevine genotype “Trincadeira” showed a tendency to have increased PE content. The down-accumulation in both lipid classes after inoculation may be connected to a further biosynthesis of lipid-related signaling molecules when the plant is under stress, since the hydrolysis of structural membrane phospholipids, such as PCs and PEs, by PLD (phospholipase D) primarily contributes to PA (phosphatidic acid) synthesis [44].

As the result of glycerophospholipids’ hydrolyzation, PA is a glycerolipid metabolic precursor as well as a signaling molecule that controls developmental, physiological, and stress responses [45]. Moreover, this is a key lipid compound in the process of defense signaling. It can cause such defense responses as ROS generation, expression of defense genes, and PCD [46]. PCD-mediated resistance is exerted inside the penetrated epidermal cell and induces the death of the invaded cell, thereby terminating the supply of nutrients required by the biotrophic fungus for further growth and development [47]. In our study, PA was found to be down-accumulated in F26P92 at 24 hpi. This is in line with [16]’s study, which found that the resistant grapevine genotype “Regent” had a higher content of PA than the susceptible genotype “Trincadeira” before being inoculated with *P. viticola*, and that the amount of PA in the resistant genotype decreased after inoculation to be comparable with that found in the susceptible genotype. This behavior could be explained by the PA biosynthesis using the slower PLD pathway rather than the faster PLC and DGK pathways [44], but further investigation is required to confirm this.

It is worth noting that the down-accumulation of the lipid classes MGDGs (monogalactosyldiacylglycerols) in “Kismish vatkhana” at 24 hpi also happened in the resistant genotype F26P92 at the same time point. Interestingly, the same class was up-accumulated by 48 hpi in “Kismish vatkhana”, while in “Teroldego” the DGDGs became up-accumulated at the same time point. Moreover, the class of DGDG was seen to be down-accumulated at 24 hpi in the resistant genotypes BC4 and F26P92 as well. Similar findings were reported by [9], who observed an increase in galactolipid levels during the incompatible interaction between grapevine and *P. viticola*. In this study, the galactolipids MGDG and DGDG were found to be substantially higher in the susceptible cultivar than in the tolerant one. This could be important in keeping cells functioning normally during a pathogen attack [16]. According to the literature, the two main lipid compounds of chloroplast membranes (MGDGs and DGDG) are required at different stages and function solely in their respective functions throughout the induction of SAR (systemic acquired resistance) and plant defenses [48]. Furthermore, MGDG is required for thylakoid synthesis in plant leaves and contributes to membrane firmness.

The behavior of the resistant genotype “Kismish vatkhana” in response to the infection with the pathogen by showing in general a lower number of down- and up-accumulated lipids than the first two resistant genotypes can be explained by the fact that *E. necator* is an adapted pathogen in this grape genotype [39]. The limited modulation noticed at 24 hpi, predominately the down-accumulation of lipid classes, suggests that *E. necator* is indeed able to enter the epidermal cells of “Kishmish vatkana” and draw nutrients from the host to sustain its initial growth [39]. The increasing modulation that we observed from 48 hpi onwards could be explained by the fact that resistance to the pathogen in “Kishmish vatkana” results in the restriction of hyphal development and a decrease in conidiophore production, which are statistically significant compared to those seen in the symptomatic controls at around 72–120 h after fungal entry [39]. The same study indicates that, nevertheless, hyphal proliferation and conidiophore density were significantly lower than in the susceptible control, which is symptomatic of PM to the unaided eye [39], thereby also confirming our phenotypic OIV-455 score assessment for this genotype.

Plants that are resistant to powdery mildews may be so as a consequence of a single defense mechanism acting alone or as a result of multiple mechanisms working together to prevent fungal development in the host. According to research, there are at least two distinct lines of defense against powdery mildews, pre-invasion and PAMPs, which prevent pathogen ingress and the onset of the pathogenic process, and ETI, which prevents further invasion if the first line of defense is overcome by pathogenic effectors [49,50,51]. Hence, the resistance mechanism in “Kishmish vatkana” is clearly at the level of the post-invasion response, as discovered by [39] and corroborated by our findings. Thus, if the pathogen seems to be able to take nutrients from its host in the first 24 h in “Kishmish vatkana”, BC4 and F26P92 appear to have a better and more restrictive defense at that time point, indicating a resistance mechanism at the pre-invasion level.

## 4. Materials and Methods

### 4.1. Plant Material

We conducted a two-year study (2019 and 2021) on three grapevine genotypes deemed resistant to *E. necator*: BC4 and “Kishmish vatkana”, each carrying one resistant locus (*Run1* and *Ren1*, respectively); the pyramided variety F26P92, carrying two resistant loci, *Ren3* and *Ren9*; and one susceptible variety, “Teroldego”.

The BC4 hybrid was developed in France and is the result of an intergeneric cross between *Muscadinia rotundifolia* and *Vitis vinifera* [52]. It is resistant to the *E. necator* pathogen via the locus *Run1*, which was one of the first *E. necator* resistance loci identified in grapevine and one of the few that has been well studied from a causal gene standpoint [20].

“Kishmish vatkana” is a cultivated grape from Central Asia created by crossing “Vasarga Chernaya” with “Sultanina” that is resistant through the *Ren1* locus [39], whereas F26P92 is a pyramided hybrid created at Fondazione Edmund Mach (Italy) from “Bianca” and “Nosiola” and carries two resistant loci, *Ren3* and *Ren9*. They are both mid-resistant genotypes. Table 1 summarizes all the resistance sources and associated resistance-related loci (*Ren* and/or *Run*) of the genotypes investigated.

### 4.2. Experimental Design and Artificial Inoculation

A total of sixty plants grafted onto Kober 5BB rootstock (n = 15 per genotype) were grown in potted soil in controlled greenhouse conditions at the Fondazione Edmund Mach located in San Michele all’Adige (Trento), Italy (46°12′0″ N, 11°8′0″ E).

Two weeks prior to the experiment, the plants were treated with sulfur to guarantee that they were pathogen-free. During the experiment, healthy plants were divided into two homogeneous groups (control and infected), and the same group of plants was further divided into three groups, each representing one biological replication (Figure 4).

The inoculation with *E. necator* was achieved according to the modified methods of [53,54], described in [23]. Briefly, naturally infected powdery mildew leaves from the same untreated vineyard of the grape variety “Pinot Noir” were collected. The inoculum, which was made of a variety of strains, was used to dust the spores with an air pump onto the adaxial surface of the healthy leaves and immediately covered with plastic bags for 24 h, while control plants were sprayed with sulfur. Following a randomization method, leaves were sampled at three time points, 0, 24, and 48 h post-inoculation/mock, immediately frozen with liquid nitrogen, and stored at −80 °C.

### 4.3. Disease Assessment

The OIV-455 descriptors scale was used to evaluate the resistance of infected leaves to the pathogen *E. necator* [36]. According to [55], a distinct plant that had been infected at the start of the experiment was subjected to a visual evaluation at 3, 7, and 14 (dpi). Generally, under constant optimum temperatures, PM can have a latent phase of 5 days until the appearance of the first visible symptoms [19,47]. Hence, in this study, we assessed the disease at 7 dpi.

### 4.4. Lipid Extraction and Analysis

Lipid extraction was carried out according to the method of [56] with some modifications. Briefly, two extractions of 100 mg of fresh leaves were collected and weighed in an Eppendorf microtube. The first fraction extraction was achieved with 0.3 mL of methanol and 0.6 mL of chloroform containing butylated hydroxyl toluene (500 mg/L), to which we added 15 µL of IS stearic acid (10 µg/mL) and 15 µL of IS, a mixture for each class of compounds (10 µg/mL), as established in [3]. The samples were then placed in an orbital shaker for 60 min; additionally, 250 µL of Milli-Q purified H_2_O was added and the extracting mixture was centrifuged for 10 min at 4 °C. For the second extracted fraction, 400 µL of CHCl3/CH3OH/H2O 86:13:1 (*v*/*v*/*v*) was used, followed by centrifugation. The combined total extract was collected in a new Eppendorf microtube and evaporated to dryness under N2. Samples were re-suspended in 300 µL of acetonitrile–2-propanol–water (65:30:5 *v*/*v*/*v*/), centrifuged at 3600 rpm at 4 °C for 5 min, and then finally transferred into HPLC vials at a volume of 250 µL. Two quantitative control (QC) samples of 100 µL each for infected and non-infected conditions were prepared using 25 µL from the pool of all sample extracts and injected in the same conditions as the individual samples.

Lipid compounds analysis was carried out according to the new method developed by [3]. The separation was performed with an Exion LC system provided by AB Sciex LLC (Framingham, MA, USA) coupled with an AB Sciex LLC QTRAP 6500+ (Framingham, MA, USA) mass spectrometer. An Acquity CSH-C18 column (2.1 × 100 mm, 1.7 µm) (Waters, Milford, MA, USA) was used in a 30 min multi-step gradient.

### 4.5. Data Processing

MultiQuant, version 3.0, was used to process the data (Sciex, Concord, Vaughan, ON, Canada). Lipid identification was validated by plotting the retention time of each compound versus its corresponding Kendrick mass defect to the hydrogen base. Lipids were semi-quantified using reference standards. Thereafter, they were corrected for the exact initial weight of leaf powder prepared during sample preparation. The number of compounds per class included in the method, the validation parameters assessed using the IS mix, the number of compounds found in our reference matrix, and the number of compounds validated are all displayed in Appendix A (sheets 1 and 2).

### 4.6. Data Analysis

A tailored R script was used for statistical analysis [57]. In order to obtain an overview of the data, we performed a principal component analysis (PCA) after applying the base 10 logarithm and UV scaling (Appendix A). The PCA indicated that the main source of variability is associated with the year, and we thus removed the year effect by subtracting the average effect of each year for each metabolite/genotype for all the following analyses.

We applied a set of univariate non-parametric tests to characterize the differential response of the distinct lipids at 24 and 48 hpi. We did not consider 0 hpi, since at that time, the plant lipidome was not expected to be different based on the infection status. To identify the lipids that were significantly altered after infection, the non-parametric Wilcoxon test was performed, followed by Cohen’s d effect size. A series of “volcano graphs” were created by combining statistical significance and effect size. To select strongly reacting lipids, uncorrected *p* < 0.05 and d > 1 were employed as arbitrary thresholds. According to [58]’s research, “d” values can range from very small (d = 0.01) to very large (d = 2.0). Appendix A lists the “d” values, associated effect sizes, and *p*-values for the found modulated lipids in all genotypes. No statistical analysis was conducted on the qualitative evaluations of leaf health (i.e., OIV-455).

## 5. Conclusions

Understanding how plants react to *E. necator* may shed some light on how plant and pathogen mechanisms have co-evolved and how that has affected plants’ resistance or susceptibility to infections. The study of plant–pathogen interactions in grapevine is crucial for understanding how pathogens attack the plant and how plant defenses are activated and strengthened. An overall picture of the lipidome changes occurring in three resistant genotypes (two mono-locus and one pyramided) versus a susceptible one in response to *E. necator* inoculation was obtained in this study using a semi-targeted lipidomics technique. Therefore, our results provide new evidence of lipids’ role in the grapevine–*E. necator* pathosystem.

In the first hours after pathogen inoculation, differential modulation of lipids was found, being more pronounced in the resistant genotypes BC4 and F26P92, and less so in “Kishmish vatkhana”. After inoculation, the resistant genotype presented an alteration in several lipid classes, mainly in the extra-plastidial lipids, in the signaling lipids, and in the plastid lipids. In the susceptible genotype, lipid modulation upon pathogen inoculation was observable at the last time point, thus suggesting that this process is activated much later than in the resistant genotypes. This could be related to an effort by the plant to establish an incompatible interaction with the pathogen. While higher levels of PCs, PEs, PGs, PAs, and PIs could be further evaluated for the identification of putative biomarkers for resistance and thus a potential resistance trait to be used in breeding programs, the DGDG and MGDG lipid classes may be highlighted as potential biomarkers for susceptibility. Further research into the biological roles of these lipids should pave the way for determining their importance in plant developmental processes and defense systems. Furthermore, examining additional time points of contact between this pathogen and grapevine will help us better understand the role of lipids in plant defense.

A thorough understanding of the function of lipid molecules and their signaling pathways in grapevine resistance mechanisms may help us define new disease control strategies by revealing the molecular mechanism underlying processes of resistance/susceptibility to fungal pathogens that in the future might help us in developing cultivar selection techniques.

## Figures and Tables

**Figure 1 ijms-24-04072-f001:**
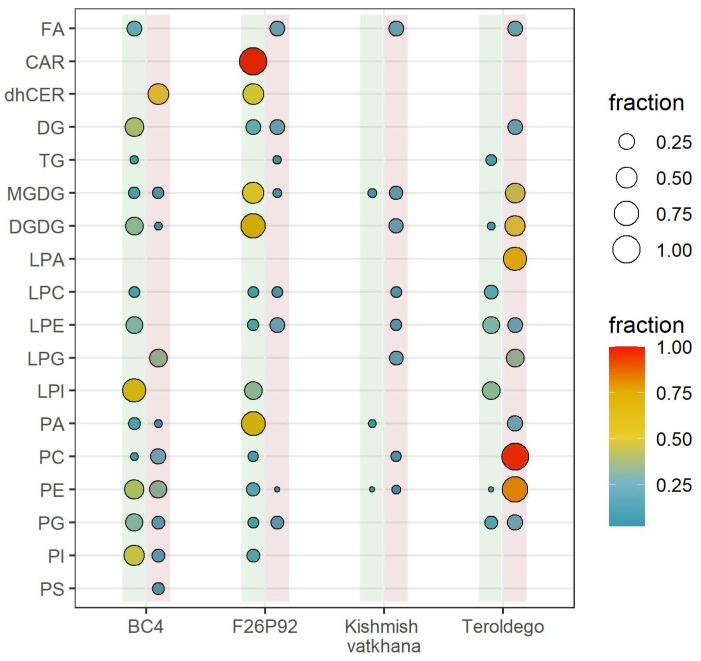
Visualization (in percentage points) of all classes of lipids that were highly modulated in response to *E. necator* infection. Based on the total number of detected and semi-quantified lipids, the size and color intensity of the dots are proportionate to the estimated percentage of lipid class modulation in each genotype in both years. The dots inside the vertical lines show the percentages of lipid modulation (green = at 24 hpi, red = at 48 hpi).

**Figure 2 ijms-24-04072-f002:**
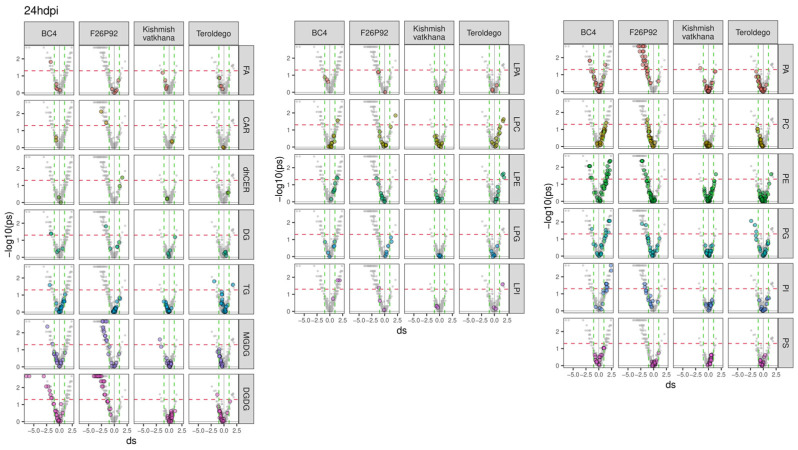
Lipids with values above the discontinued red line were significantly modulated with the up-accumulated lipids shown on the right and down-accumulated lipids shown on the left arm of the volcano for all four genotypes at 24 hpi over the course of the two years of data analysis (2019–2021). The left graph shows the modulation of glycerolipids, sphingolipids, and fatty acids, whereas the middle and right graphs show the modulation of glycerophospholipids. The colors reflect the various lipid classes, while “ds” represents the calculated Cohen’s d values.

**Figure 3 ijms-24-04072-f003:**
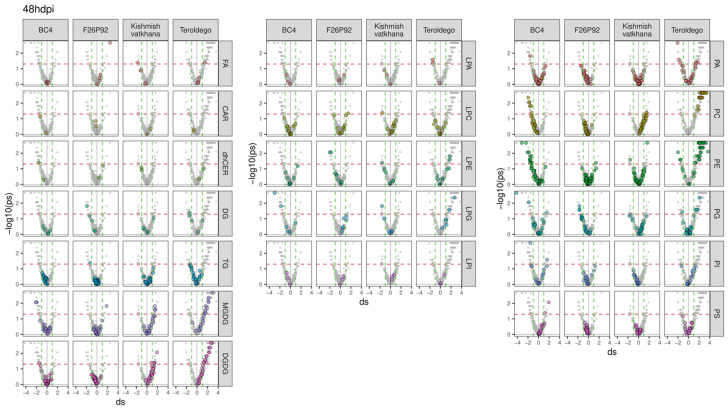
Lipids with values above the discontinued red line were significantly modulated, with up-accumulated lipids shown on the right and down-accumulated lipids shown on the left of the volcano plot for all four genotypes at 48 hpi over the course of the two years of data analysis (2019–2021). The left graph shows the modulation of glycerolipids, sphingolipids, and fatty acids, whereas the middle and right graphs show the modulation of glycerophospholipids. The colors reflect the various lipid classes, while “ds” represents the calculated Cohen’s d values.

**Figure 4 ijms-24-04072-f004:**
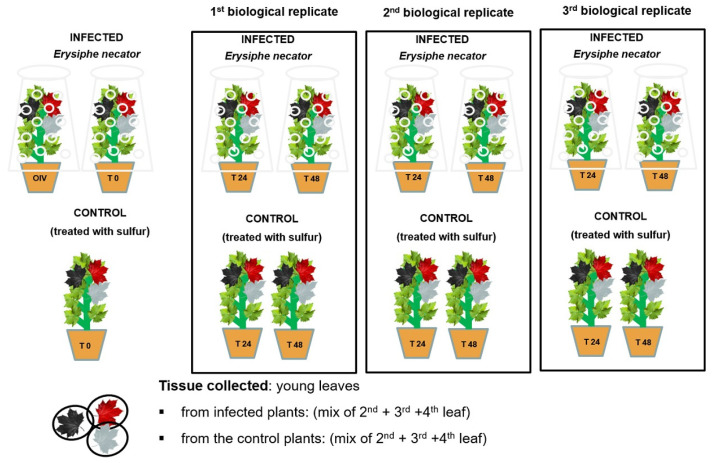
Randomization scheme of *E. necator’*s inoculation and sample collection. The graph shows the three biological replicates, each with three time points (0, 24, and 48 hpi). Each biological replicate was divided in two groups: infected and control. The sample material collected was the second, third, and fourth leaf taken from each time point within each biological replicate, whereas the control was a mixture of the second, third, and fourth leaf taken from all the plants in a biological replicate.

**Table 1 ijms-24-04072-t001:** The grapevine varieties used in this study together with their origin (^1^ North American *Vitis*; ^2^ pure *V. vinifera*, ^3^ interspecific hybrids of *V. vinifera* with North American *Vitis* species), host response (PCD (programmed cell death), ROSs (reactive oxygen species), n.d. (not determined)), and their powdery-mildew-associated resistance-related loci (*Ren*/*Run*). The levels of resistance described in the table: total = greatly suppressed symptoms or the absence of visible symptoms; partial = in cases where the symptomatology decreases without disappearing completely [35,36].

Genotypes		Resistance- Related Powdery Mildew Loci (*Ren*/*Run*)	Resistance Mechanism within the Hosts			Preliminary Leaf Resistance Level	Source of Resistance	References
			PCD	ROS	Callose			
mono-locus resistance	BC4	*Run1*	yes	yes	yes	total resistance	*M. rotundifolia* ^1^	[20,37]
	“Kishmish vatkana”	*Ren1*	yes	yes	yes	partial resistance	*V. vinifera* ^2^	[39]
pyramided resistance	F26P92	*Ren3*	yes	yes	yes	partial resistance	*V. rupestris* ^3^	[34,38]
		*Ren9*	yes	n.d.	n.d.	partial resistance	*V. rupestris* ^3^	[34,40]
control	“Teroldego”	-	-	-	-	susceptible	-	

## Data Availability

The original contributions presented in the study are included in the article/Appendix A. Further inquiries can be directed to the corresponding author.

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
