# Peer review of "Semi-Targeted Profiling of the Lipidome Changes Induced by Erysiphe Necator in Disease-Resistant and Vitis vinifera L. Varieties"

_ijms, 2023, doi:10.3390/ijms24044072_

Round 1

Reviewer 1 Report

Although paper seems very fascinating, for usual reader, I recommend to do the abbreviations' list at the beginning. Earlier in literature, main attention paid to volatile essential oils of the species  as a resistance factor against any infection.  Are lipids  that you studied more effective in a comparison with essential oils?

Reviewer 2 Report

This is a manuscript review for Semi-targeted profiling of the lipidome changes induced by Erysiphe necator in disease-resistant and Vitis vinifera L. varieties.

The article added important new knowledge on lipidome changes. The article is well written. There are a few minor collections.

Table 1: Resistance-related powdery mildew loci (Ren/Run) should be in italics if they are genes. Check the rest of the manuscript.

Table 1: Does n.d. mean not determined? Add a footnote or description.

Fig 4: A bit more description of the graphics should be added in the legend.

Supplementary figures need renumbering for clarity.

Supplementary Figure 1: There is no legend accompanying the figure

Supplementary Table 3: When the file is opened it shows Table 3. Add 'S' or 'supplementary' before 'S'

Supplementary Table 2: The file name shows ' Supplementary Table 2' but when opened, Supplementary Table 1 is seen. Revise.

Supplementary Table 1: The file name is 'Supplementary Table 1' but when opened, the first worksheet shows Supplementary Table 4, The next sheet is Table S3 and the next sheet has no legend.
